# Proximal and distant expression of growth differentiation factor 15 (GDF15) correlate with neurological deficit following experimental ischemic stroke

Alexandre Méloux[1,2], Geoffrey Dogon[1], Eve Rigal[1], Luc Rochette[1], Yannick Bejot[1,3], Catherine Vergely [1] *

1 Physiopathologie et Epidémiologie Cérébro-Cardiovasculaires (PEC2), Faculty of Health Sciences, Université de Bourgogne, Dijon, France, 2 UMR INSERM 1093, Cognition, Action et Plasticité Sensorimotrice, Université de Bourgogne, Dijon, France, 3 Department of Neurology, University Hospital of Dijon, Dijon, France

* cvergely@u-bourgogne.fr

## Abstract

### Background and purpose

Growth differentiation factor 15 (GDF15) has emerged as a promising biomarker in cerebro-cardiovascular disease, particularly in acute and chronic inflammatory stress situations. However, understanding the origins, targets and functions of GDF15 in clinical situations, such as ischemic stroke, remains a complex challenge. This study aims to assess the sources of GDF15 production following an experimental ischemic stroke.

### Methods

Adult male Wistar rats underwent cerebral embolization through microspheres injection into the left or right internal carotid artery. Two hours post-surgery, GDF15 expression was analyzed in the brain, blood, lungs, liver and heart using quantitative RT-PCR and Western blotting.

### Results

Stroke model induced large cerebral infarcts accompanied by severe neurological deficits. GDF15 gene expression exhibited a substantial increase in the ipsilateral cortex and cerebellum, with a lesser extent in the contralateral cortex. Regarding GDF15 protein expression, proGDF15 levels were elevated in the 3 aforementioned organs mentioned and the heart. However, the mature form of GDF15 was exclusively present and increased in the heart. Finally, the expression of GDF15 expression was correlated with the neurological deficit score.

### Conclusions

Our findings suggest that both the GDF15 gene and pro-protein are expressed in the ischemic brain after a stroke, while only its mature form is expressed remotely in in the heart. The

**Data Availability Statement:** "***PA AT ACCEPT: Please follow up with authors for data available at

**Funding:** This study has been supported by funding from the French Ministry of Research (CV), from the Regional Council of Burgundy (CV, YB), from the Association Bourguignonne de Cardiologie, and from the Regional University Hospital (YB, AM) and Faculty of Health Sciences (ER, GD) and from the ANR (SMOG15-CE17-009-01, YB, CV). The funders had no role in study design, data collection and analysis, decision to publish, or preparation of the manuscript.

**Competing interests:** The authors have declared that no competing interests exist.

impact of increased GDF15 in the heart following a stroke remains to be established. This is particularly relevant in understanding its relationships with poor neurological outcomes, determining whether it may contribute to stroke-induced cardiac dysfunction.

## Introduction

Growth differentiation factor 15 (GDF15), a distant member of the superfamily of transforming growth factor-β which receptor has recently been discovered [1], is gaining recognition as a biomarker in cardiovascular diseases (CV) and a predictor of all-cause mortality [2]. This protein is initially secreted as a precursor form which, after proteolysis, is transformed into proGDF15, subsequently undergoing cleavage to yield mature GDF15. Under physiological conditions, GDF15 is expressed at very low levels, but its circulating levels surge in certain pathological situations, related to inflammatory stress and aging.

Among these pathological situations, atherosclerosis is a notable example. Indeed, GDF15 appears to be highly overexpressed in macrophages, playing a potential role in orchestrating atherosclerotic lesions by triggering a proinflammatory response and a proapoptotic effect [3]. In mice, GDF15 deficiency has been linked to the reduction in the development of atherosclerotic lesions, achieved through the modulation of IL-6-dependent inflammatory response [4]. Additionally, GDF15 has demonstrated influence over the accumulation of lipids and autophagy proteins, highlighting a pivotal role in the pathophysiology of atherosclerotic plaque development and progression [5]. It is crucial to emphasize that atherosclerosis stand as one of the etiological factors contributing to stroke. GDF15 levels are thus associated with the occurrence of a transient ischemic attack and even an ischemic stroke (IS) after adjustment for age and sex [6]. After an experimental IS, circulating GDF15 increases very quickly [7]. In clinical settings, GDF15 very high levels [7, 8] are observed in IS patients upon admission, correlating with poor functional outcomes [9] and increased mortality [8, 10].

To comprehensively understand the targets and functions of GDF15 in clinical situations, it is crucial to unravel its tissue and cellular sources post-IS. This is particularly significant considering the potential transmission of detrimental signals from the ischemic brain to distant organs such as the heart [11, 12]. Therefore, the aim of this study was to elucidate the origins of GDF15 following an experimental IS.

## Methods

This article adheres to the AHA Journals implementation of the Transparency and Openness Promotion Guidelines.

### Animals

Adult male Wistar rats (Charles River, Ecully, France) (8–9 weeks, 300 ± 50 g) were utilized. All animals received humane care, and experimental protocols adhered to institutional guidelines. The investigation was conducted in accordance with Directive 2010/63/EU of the European Parliament and to the Guide for the Care and Use of Laboratory Animals published by the US National Institutes of Health (NIH Publication No. 85–23, revised 1996). Approval was obtained from the local ethics committee (protocol agreement number: APAFIS#16546). Animals were provided with *ad libitum* access to standard diet and water.

## Model of cerebral embolization and neurological deficit assessment

As previously outlined [7], left or right cerebral embolization was conducted with microspheres. Briefly, rats were deeply anesthetized with 3.5% isoflurane in $O_2$ and local analgesia with lidocaine/prilocaine was applied on the shaved skin of the neck to prevent surgical pain. After dissection of the common carotid artery and ligation of the external carotid artery, microspheres were injected into the common carotid blood flow. This embolization methods caused multiple, disseminated infarcts that mimicked a cardioembolic stoke. Just after the surgical procedure, a buprenorphine injection (0.05 mg/kg, s.c., Buprécare, Axience, Pantin, France) was carried out in order to limit the animals' suffering. Neurological deficit was assessed 2 h post-embolization employing the modified Longa score [13] by a blinded observer.

## Quantitative real-time polymerase chain reaction

Two hours after cerebral embolization, euthanasia was induced through intraperitoneal injection of 110 mg/kg sodium pentobarbital. Heart, lung and liver tissues were briefly rinsed in saline and blotted on paper before freezing in liquid nitrogen. The brain and cerebellum were also harvested with minimal pre-freezing procedures. As previously detailed [7], a semiquantitative analysis of GDF15 were conducted on various tissues including total blood, brainstem, cerebellum, contralateral cortex, ipsilateral cortex, heart, liver, lung and muscle. Total RNA was extracted by Qiazol reagent (Invitrogen, Life Technologies, Saint Aubin, France). Final RNA quality and quantity were verified by agarose gel electrophoresis (Agilent2100 bioanalyzer, Agilent technologies, Waldbronn, Germany) and measured using a spectrophotometer (Nanodrop 1000, Thermo Fisher scientific, Wilmington, USA). The specific primers utilized, sourced from Invitrogen were as follows: GDF15 F: 5′-CGAGAGGACTCGAACTCAGA-3′; R: 5′-CCCAATCTCACCTCTGGACT-3′; GAPDH F: 5′-CTACCCACGGCAAGTTCAAC-3′; R: 5′-CCAGTAGACTCCACGACATAC-3′. Each sample was measured in triplicate and relative mRNA expression was calculated using the following equation: $2^{-\Delta\Delta CT}$.

## Western blotting

As previously outlined [7], protein expressions were assessed trough Western blotting. Briefly, tissue was homogenized in 10 volumes of radioimmunoprecipitation assay (RIPA) buffer and protein concentration in the supernatant was determined using the Lowry method. Equal protein amounts were separated using SDS-polyacrylamide gel electrophoresis and transferred to a 0.45 μm polyvinylidene difluoride membrane. Primary antibodies directed against GDF15 (1:1,000, Abcam, Cambridge, UK), HPRT (as internal control, 1:10,000, Abcam) and anti-rabbit (1:1,000, Cell Signaling Technology) IgG horseradish peroxidase-linked antibodies were employed. The bands were visualized using Clarity western ECL substrate chemiluminescent detection reagent (Bio-Rad, Hercules, USA). The bands were determined with the ChemiDoc Imaging System (Bio-Rad) and analyzed with Image Lab software version 6.1 (Bio-Rad).

## Determination of plasma GDF15

Blood was collected from the abdominal aorta, and immediately centrifuged at 4°C to separate the plasma, and samples were stored at -80°C. Plasma levels of GDF15 were measured using a commercially available ELISA kit (MGD150, R&D systems, MN). All assays were performed in duplicate.

## Statistics

Data are presented as mean±SEM. The statistically significant differences were tested by analysis of variance (ANOVA) or Kruskal-Wallis test depending on the data normality. Pearson or Spearman correlations were conducted depending on the data normality. The difference was considered significant when $P < 0.05$.

## Results

Following cerebral embolization, stroke rats exhibited significantly severe neurological deficit scores compared to the sham group (see S1 Fig).

### Cerebral embolization induces a notable upregulation of GDF15 gene expression in the brain

Two hours after IS, GDF15 gene expression was evaluated in various tissues. Cerebral embolization, whether on the left or right, resulted in more than ten-fold increase in GDF15 mRNA expression in the ipsilateral cortex and the cerebellum. Additionally, a 3-fold increase in GDF15 gene expression was noted in the contralateral cortex, reaching statistical significance only in the right stroke group (Fig 1). Notably, no elevation in GDF15 mRNA expression was observed in organs distant from the brain, such as the heart, or in circulating blood cells (S2 Fig).

### Cerebral embolization induces an elevation in the expression of proGDF15 protein in both the brain and the heart, while mature GDF15 is exclusively increased in the heart

The expression of GDF15 protein was examined in the same tissues. The results revealed an approximately 50% increase in the expression of proGDF15 protein, the precursor of GDF15, in both the ipsilateral and contralateral cortex, the cerebellum and the heart after left and right cerebral embolization. The mature form of GDF15 was exclusively detected in the heart, displaying a significant 75% increase in both left and right stroke groups compared to the sham group (Fig 2). Corresponding uncropped membranes are available in supporting information (S3 Fig). Importantly, no discernable in GDF15 mature protein expression was observed in others organs.

### The expression GDF15 is correlated with neurological deficit score

From a genetic standpoint, the expression of the GDF15 gene in the ipsilateral, contralateral and cerebellum regions exhibited a positive correlation with the neurological deficit score (Fig 3). Similar correlations were observed for proGDF15 protein expression, not only in these brain regions but also in the heart. Moreover, the mature form of GDF15 showed correlation with the neurological deficit score (Fig 4). When considering previously published data on circulating GDF15, a correlation between circulating GDF15 and the neurological deficit score was evident at 24 h post-embolization (S4 Fig).

## Discussion

This study reveals, for the first time, an elevation in GDF15 expression following experimental IS, not only in the ischemic brain and nearby regions such as the contralateral cortex and cerebellum, but also distantly, in a remote organ: the heart. Moreover, our results reveal a

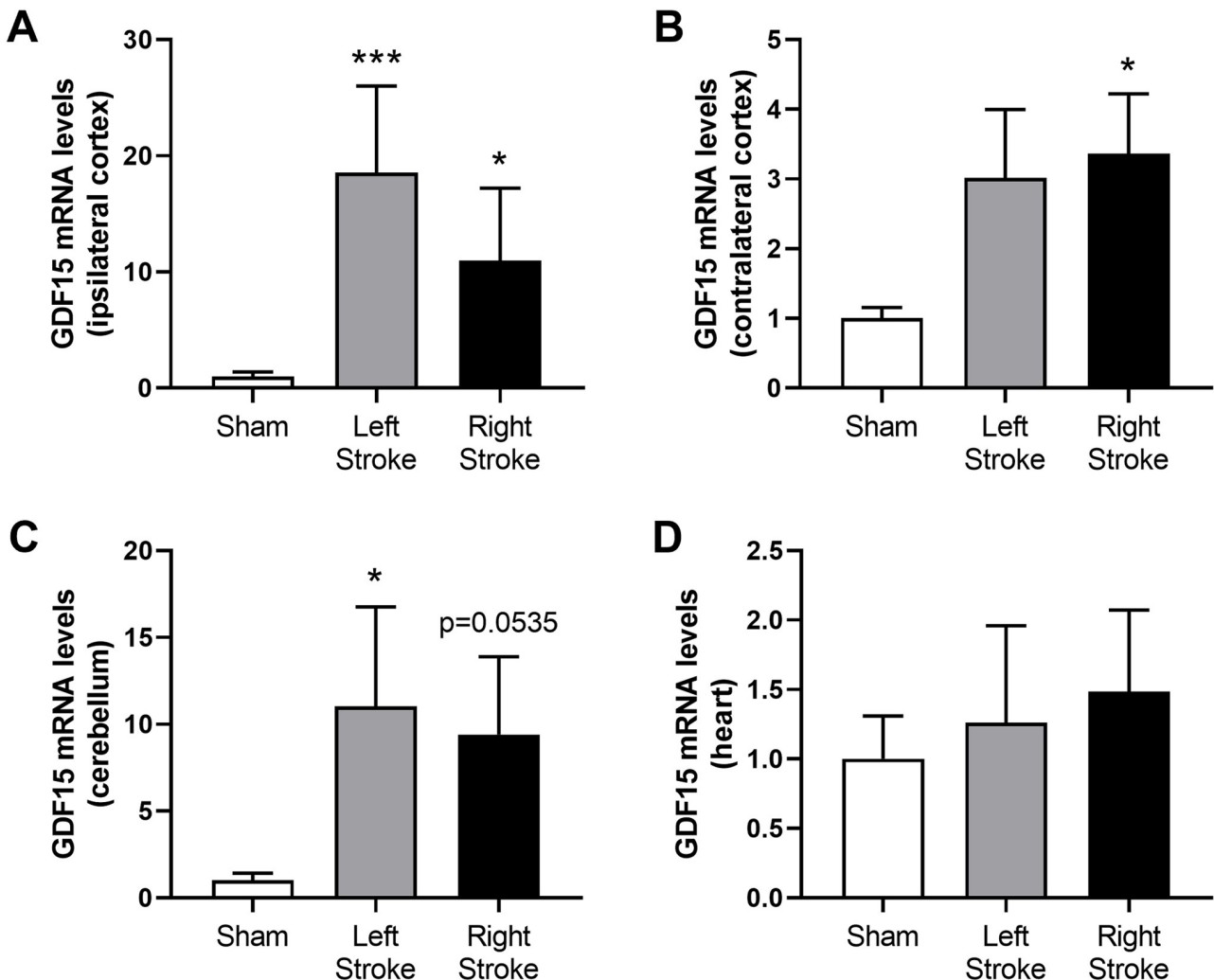

**Fig 1. Effect of cerebral embolization on GDF15 mRNA expression in ipsilateral cortex (A), contralateral cortex (B), cerebellum (C) and heart (D) 2 h after embolization or sham surgery, n = 8 per group.** * p<0.05 and *** p<0.001 compared to Sham group.

correlation between the level of gene or protein expression of GDF15 in various tissues and the neurological deficit following experimental IS.

Our experimental data support previous observation that GDF15 circulating levels rapidly increase in the circulation following an experimental IS [7], aligning with the elevated levels of GDF15 observed in stroke patients [10]. Notably, GDF15 may originate from various tissues, including the ischemic brain, particularly from inflammatory cells. Our experimental results suggest that circulating GDF15 primary emanates from ischemic brain areas (cortex and cerebellum). Just 2 h after cerebral embolization, a significant 10-fold increase in GDF15 mRNA expression was observed and, to a lesser extent (1.5-fold), the pro-protein. These findings are consistent with earlier results demonstrating increased GDF15 gene expression after middle cerebral artery occlusion (MCAO) in the hippocampus and ipsilateral cortex [14]. However, contrary to others observations [15], GDF15 gene expression was also moderately increased in the contralateral cortex of right stroke rats. This unexpected result could be attributed to the embolization model employed, inducing multiple small infarctions and a general

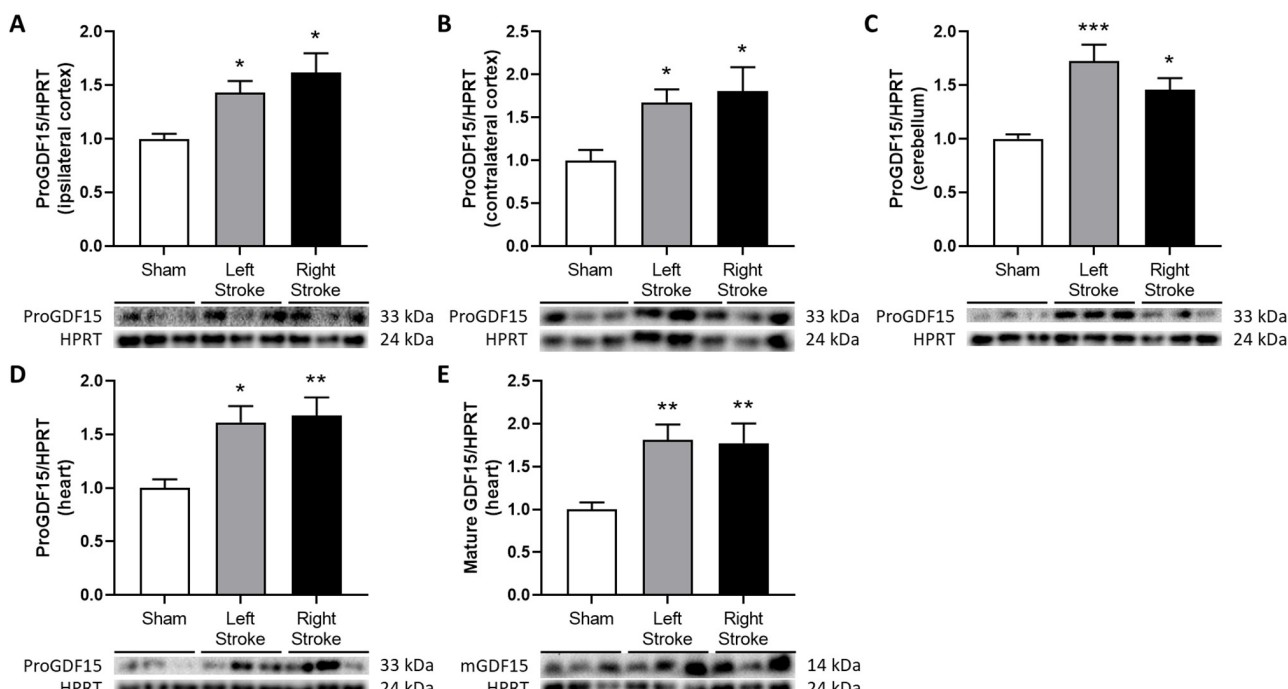

**Fig 2. ProGDF15 and mature GDF15 relative expression.** ProGDF15 in ipsilateral cortex (A), contralateral cortex (B), cerebellum (C) and heart (D) and mature GDF15 in heart (E) determined by Western blot analysis 2 h after embolization. Data were normalized to HPRT, n = 4–10 per group. * p<0.05, ** p< 0.01 and *** p<0.001 compared to Sham group.

inflammation in the brain that may extend to the contralateral cortex. According to our previous data, [7] some of the microspheres inadvertently reaches the contralateral cortex via the Circle of Willis. Therefore, the observed elevation of GDF15 in these brain regions might be attributable to small embolic strokes occurring in this area. Previous studies using immunohistochemistry or ELISA for GDF15 protein expression in the ischemic zone, cannot distinguish between the pro- and the mature forms of the protein. The novelty of this work lies in the discovery of a very early activation of GDF15 gene and of non-mature pro-protein expression, not only in the brain, but also in distant organs such as the heart. Interestingly, highly vascularized organs such as the liver, lungs and muscles, did not show an increase in GDF15 protein expression, eliminating the hypothesis of a direct link between elevated tissue and circulating blood GDF15. Indeed, the mature form of GDF15 was only detected in cardiac tissue, where mRNA expression was not increased, suggesting a predominantly post-translational activation process. The consequences of remotely-induced GDF15 upregulation in the heart by cerebral ischemia are unknown, but may reflect a new interaction between the brain and the heart during ischemic processes [12]. Current knowledge associates GDF15 with potential cardioprotection [16]; however, its role in stroke-induced cardiac dysfunction remains to be determined, and whether it constitutes a deleterious or cardioprotective factor in this context [17]. In a parallel pathological context, recent data suggest that chronic inflammation from IS can extend from the ischemic cerebellum to distant peripheral organs such as the heart [18].

A particularly innovative finding from our study is the correlation observed between the neurological deficit following experimental IS and the level of gene or protein expression of GDF15, not only in the infarcted area of the brain, but also in the contralateral hemisphere, the cerebellum, and even the heart. These findings collectively suggest that GDF15 may serve

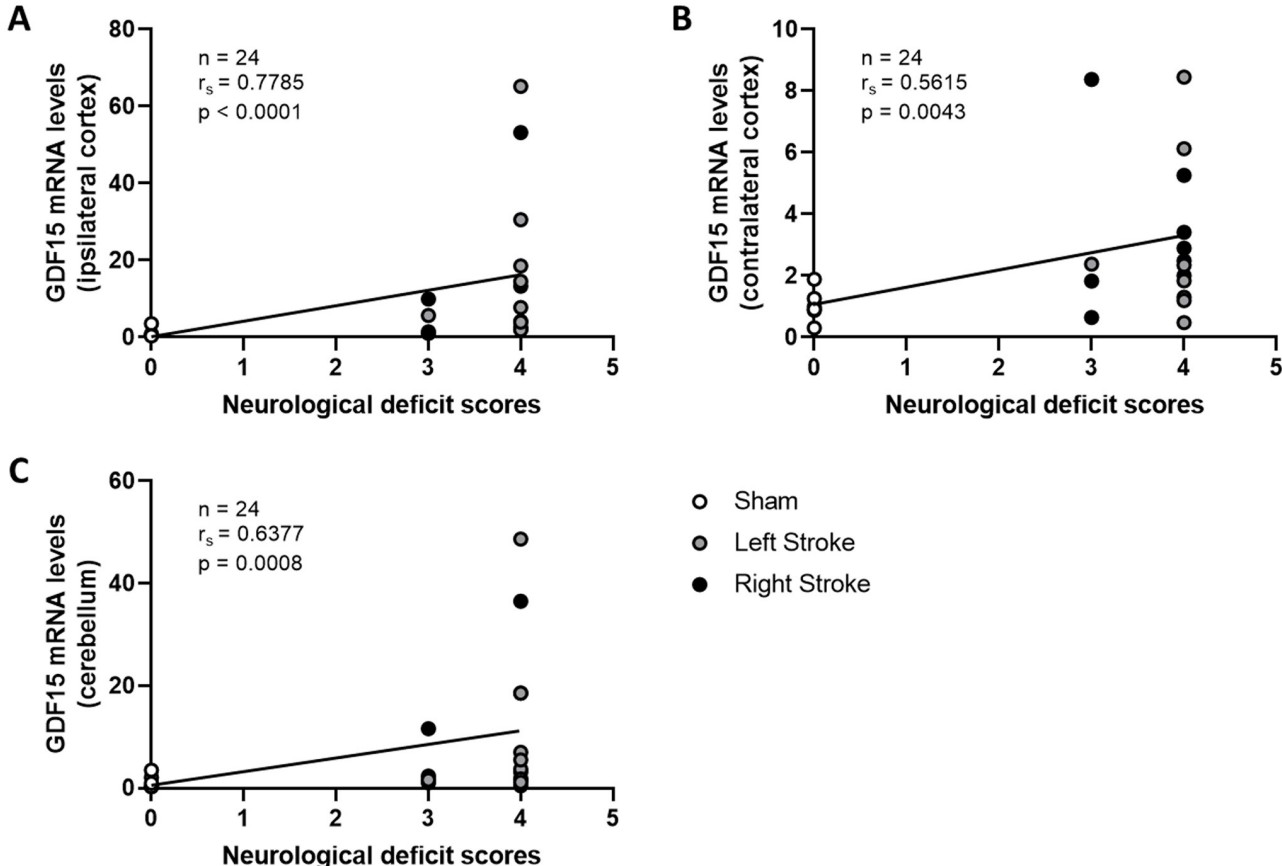

**Fig 3. Correlation between GDF15 gene expression and neurological deficit score in ipsilateral cortex (A), contralateral cortex (B) and cerebellum (C) 2 h after embolization or sham surgery, n = 8 per group.**

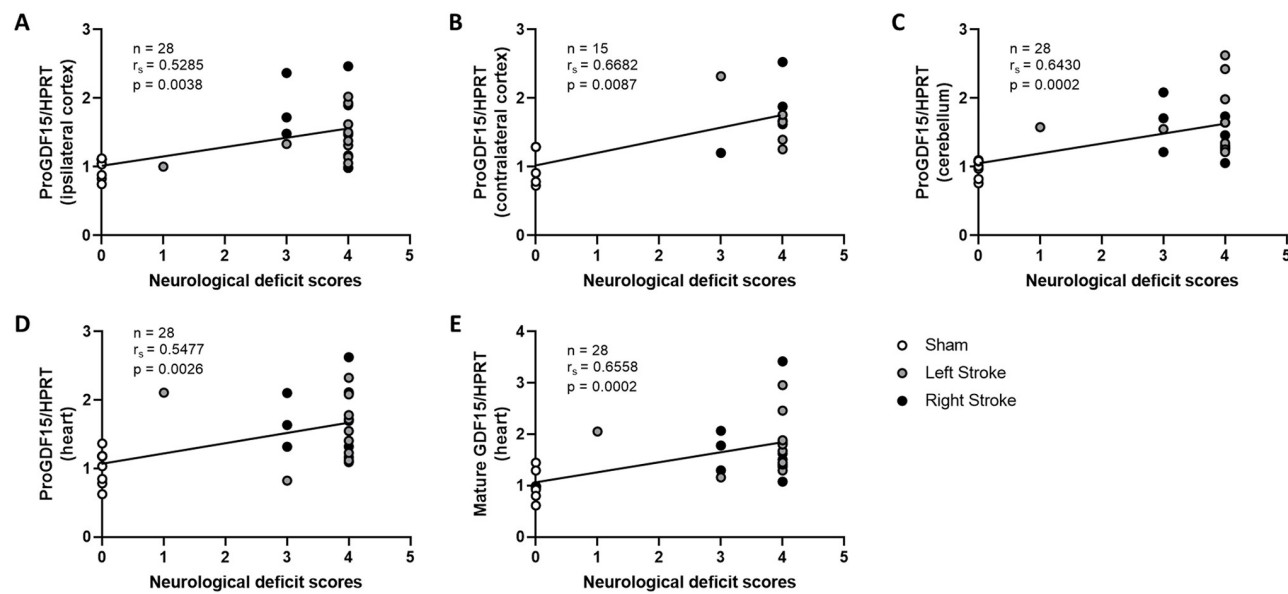

**Fig 4. Correlation between ProGDF15 protein expression and neurological deficit score in ipsilateral cortex (A), contralateral cortex (B), cerebellum (C) and heart (D) and mature GDF15 in the heart (E) 2 h after embolization or sham surgery, n = 4–8 per group.**

as a marker for neurological damages. To the best of our knowledge, there have been no prior reports documenting an association between the up-regulation of GDF15 in tissues and post-stroke functional impairment. In Japanese outpatients with cardiovascular risk factors, GDF-15 was associated with an increased risk of stroke events beyond conventional risk factors and other prognostic markers [19]. It has been reported earlier in patients with IS that circulating levels of GDF-15 are elevated and associated with neurological outcomes [20]. A previous publication from our team demonstrated that the plasma concentration of GDF-15 at admission was independently associated with 3-month mortality in IS patients undergoing acute revascularization therapy [10]. Recent data corroborate these results, exhibiting that elevated serum GDF-15 levels are associated with death and vascular events within 3 months after IS in patients with diabetes [21]. It remains unclear whether GDF15 is solely a biomarker associated with high cardio-and cerebrovascular risk, along with increased all-cause morbidity and mortality, or if it plays a significant pathophysiological role in these conditions. Further experimental studies are required to clearly identify not only the peripheral membrane receptors for GDF15 but also its intracellular molecular pathways within the cardiovascular system. The results of this work may enhance our understanding of the physiological functions of GDF15, its involvement in cerebrovascular pathologies and its relationship with poor outcomes.

## Conclusions

Our research reveals a surge in GDF15 expression following an experimental IS, both in the brain proximal to the lesion and remotely, with an intriguingly significant upregulation observed in a distant organ such as the heart. Furthermore, the correlation of elevated GDF15 expression in various tissues, including the brain and the heart, with poor neurological outcomes is an original and striking result. While our findings shed light on GDF15 post-stroke expression pattern, its intricate cardiac functions in the aftermath of a stroke are yet to be fully elucidated. The influence of GDF15 on cardiac function may extend beyond its role as a systemic inflammatory and stress marker, potentially influencing complex brain-heart interactions.

## Supporting information

**S1 Fig. Neurological deficit scores 2 h after cerebral embolization (n = 28).**
(TIF)

**S2 Fig. Effect of cerebral embolization on GDF15 mRNA expression in liver (A), lung (B), muscle (C) brainstem (D) and blood 2 h after embolization or sham surgery, n = 8 per group.**
(TIF)

**S3 Fig. Uncropped full-length pictures of western blotting membranes.** ProGDF15 in ipsi-lateral cortex (A), contralateral cortex (B), cerebellum (C) and heart (D) and mature GDF15 in heart (E) 2 h after embolization. Data were normalized to HPRT.
(TIF)

**S4 Fig. Correlation between GDF15 serum expression and neurological deficit score 24 h after embolization or sham surgery, n = 10–12 per group.**
(TIF)

## Acknowledgments

The authors thank Ivan Porcherot, Océane Gruson and Maxime Velin for technical assistance.

## Author Contributions

**Conceptualization:** Yannick Bejot, Catherine Vergely.

**Data curation:** Alexandre Méloux, Eve Rigal.

**Funding acquisition:** Yannick Bejot, Catherine Vergely.

**Investigation:** Alexandre Méloux, Geoffrey Dogon.

**Methodology:** Eve Rigal.

**Project administration:** Catherine Vergely.

**Supervision:** Catherine Vergely.

**Validation:** Luc Rochette, Catherine Vergely.

**Writing – original draft:** Alexandre Méloux.

**Writing – review & editing:** Yannick Bejot, Catherine Vergely.

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
