## [Decision Letter · Decision Letter 0]

21 May 2024

PONE-D-24-07402Proximal and distant expression of growth differentiation factor 15 (GDF15) correlate with neurological deficit following experimental ischemic strokePLOS ONE

Dear Dr. Vergely,

Thank you for submitting your manuscript to PLOS ONE. After careful consideration, we feel that it has merit but does not fully meet PLOS ONE’s publication criteria as it currently stands. Therefore, we invite you to submit a revised version of the manuscript that addresses the points raised during the review process.

This is an interesting manuscript. I agree with the reviewers that it does have a number of limitations, however. The Major Comments of reviewer 1 in particular need to be dealt with robustly as part of a revised manuscript before it could be considered suitable for publication. The other comments also require consideration, but are of slightly less importance.

We look forward to receiving your revised manuscript.

Kind regards,

Clive J. Petry, PhD

Academic Editor

PLOS ONE

“This study has been supported by funding from the French Ministry of Research (CV), from the Regional Council of Burgundy (CV, YB), from the Association Bourguignonne de Cardiologie, and from the Regional University Hospital (YB, AM) and Faculty of Health Sciences (ER, GD) and from the ANR (SMOG15-CE17-009-01, YB, CV).”

6. We notice that your supplementary figures are uploaded with the file type 'Figure'. Please amend the file type to 'Supporting Information'. Please ensure that each Supporting Information file has a legend listed in the manuscript after the references list.

Reviewers' comments:

Reviewer's Responses to Questions

**Comments to the Author**

1. Is the manuscript technically sound, and do the data support the conclusions?

Reviewer #1: Partly

Reviewer #2: Yes

2. Has the statistical analysis been performed appropriately and rigorously? 

Reviewer #1: I Don't Know

Reviewer #2: Yes

3. Have the authors made all data underlying the findings in their manuscript fully available?

Reviewer #1: Yes

Reviewer #2: Yes

4. Is the manuscript presented in an intelligible fashion and written in standard English?

Reviewer #1: Yes

Reviewer #2: Yes

5. Review Comments to the Author

Reviewer #1: This work follows up on previous work published in the journal Stroke, where the same group reported elevated levels of plasma GDF15 post-ischemic stroke (IS). The aim of the current manuscript is to identify the tissue sources of GDF15 following IS and to explore the correlation between GDF15 mRNA levels and neurological deficits associated with IS.

Major Comments:

1. The manuscript establishes that GDF15 mRNA levels increase in both ipsilateral and contralateral regions of the brain post-IS. Given that GDF15 is well documented as an early cellular stress protein, its elevation in the ipsilaterally affected areas is anticipated. However, the elevation in the contralateral brain regions prompts a question. Can the authors definitively rule out that the microsphere methodology might have induced ischemic conditions or blood flow disruptions in the contralateral hemisphere as well?

2. The western blot (WB) images provided are of suboptimal quality and, given the semi-quantitative nature of WB, it would be advantageous to consider immunoassays for such analysis. While they cannot distinguish between the pro and mature form, Immunoassays offer more precise quantification and a broader dynamic range, which could significantly strengthen the results presented.

3. The paper discusses the significance of elevated GDF15 pro-form without corresponding evidence for the mature form. This observation might be attributable to the temporal aspects of protein production and maturation. Analyzing samples at intervals beyond the 2-hour mark post-IS could provide insights into the presence and levels of the mature GDF15 form. It is also recommended that both pro and mature forms of GDF15 be presented on the same WB for comparison.

4. There is no mention of whether animals were perfused to remove excess blood before tissue analysis. The residual blood could be a confounding factor, particularly noticeable in the differential results reported for heart tissue. Please clarify this methodological aspect.

5. The use of different animals in the neurological scoring presented in Figures 3, 4, and supplementary Figure 3 raises questions about the consistency of the analysis. Could you provide a rationale for not including all animals in each set of analyses?

Minor Comments:

• The Methods section is overly concise, which makes it challenging to comprehend the exact procedures undertaken, especially concerning the preparation of animals and tissues before WB analysis. A more detailed description would be beneficial.

• The manuscript lacks a detailed method for quantifying GDF15 in plasma.

• The statistical significance denoted by one to three stars is not explained in the legend or methods. Please ensure that all symbols used in figures are adequately defined.

• The term "very significant" used in the conclusions is ambiguous. To maintain clarity and precision in reporting results, it is advisable to use established terms such as "significant." Please specify which data the term refers to and consider revising it accordingly.

Reviewer #2: The authors have carried out interesting work looking at GDF15 in the context of stroke in a pre-clinical model. This work is interesting, and novel but some additions would be useful to improve the quality of the manuscript. If the authors have brain sections available, it would be useful to complete the analyzes for immunochemistry for GDF15 between the different areas. In addition if the authors could test the detection of GFRAL in the brain cortex and discuss this results if GRAL present or not could be interesting.

6. PLOS authors have the option to publish the peer review history of their article (what does this mean?). If published, this will include your full peer review and any attached files.

Reviewer #1: No

Reviewer #2: No

---

## [Author Response · Author response to Decision Letter 0]

21 Jun 2024

Journal requirements

Thank you for considering our article for publication in PLoS One. We appreciate the opportunity to address the reviewer's queries and are committed to providing thorough and accurate responses.

1. We have adapted our manuscript in order to meet PLOS ONE’s style requirement, including those for file naming.

2. In our Methods section, we provided additional information regarding the experiments involving animals and ensured we included details on methods of sacrifice, anesthesia/analgesia and efforts to alleviated suffering.

3. In our funding sources section, we stated that the funders had no role, by including: "The funders had no role in study design, data collection and analysis, decision to publish, or preparation of the manuscript." This was also included in the cover letter.

4. We confirm that that we will make your data available on request following the acceptance of our article.

5. We provided original uncropped and unadjusted images of our blot/ in Supporting Information. This was also included in the cover letter.

6. Our supplementary figures are uploaded with the file type 'Supporting Information'. Each Supporting Information file has a legend listed in the manuscript after the references list.

 

Reviewer #1

The authors express their gratitude to the reviewer for the meticulous review of our manuscript and for the insightful comments provided.

Major Comments:

1. The manuscript establishes that GDF15 mRNA levels increase in both ipsilateral and contralateral regions of the brain post-IS. Given that GDF15 is well documented as an early cellular stress protein, its elevation in the ipsilaterally affected areas is anticipated. However, the elevation in the contralateral brain regions prompts a question. Can the authors definitively rule out that the microsphere methodology might have induced ischemic conditions or blood flow disruptions in the contralateral hemisphere as well?

We appreciate the reviewer’s concern regarding the expression of GDF15 in the contralateral cortex. According to our previous data, some of the microspheres inadvertently reaches the contralateral cortex via the Circle of Willis. Therefore, the observed elevation of GDF15 in these brain regions might be attributable to small embolic strokes occurring in this area. This caution has been addressed in the discussion section.

2. The western blot (WB) images provided are of suboptimal quality and, given the semi-quantitative nature of WB, it would be advantageous to consider immunoassays for such analysis. While they cannot distinguish between the pro and mature form, Immunoassays offer more precise quantification and a broader dynamic range, which could significantly strengthen the results presented.

We fully agree with the reviewer’s insightful comment that immunoassays would provide valuable data for accurately determining the specific brain areas involved in GDF15 expression in both parts of the cortex following stroke. Unfortunately, all brain samples used for biochemical analysis have been exhausted, and no further experiments can be conducted in the near future. We sincerely apologize for this limitation.

3. The paper discusses the significance of elevated GDF15 pro-form without corresponding evidence for the mature form. This observation might be attributable to the temporal aspects of protein production and maturation. Analyzing samples at intervals beyond the 2-hour mark post-IS could provide insights into the presence and levels of the mature GDF15 form. It is also recommended that both pro and mature forms of GDF15 be presented on the same WB for comparison.

We agree with the reviewer that gene activation, protein expression and maturation, followed by downstream cellular signaling activation is a complex process, involving both spatial and temporal aspects. We would have been enthusiastic to evaluate the fate of GDF15 in the brain and in the heart in a broader time span. However, due to the high post-stroke mortality, we decided to focus our study on the early post-stroke stages.

4. There is no mention of whether animals were perfused to remove excess blood before tissue analysis. The residual blood could be a confounding factor, particularly noticeable in the differential results reported for heart tissue. Please clarify this methodological aspect.

We thank the reviewer for bringing this point to our attention. While the heart tissue was briefly rinsed in saline and blotted on paper before freezing in liquid nitrogen, we did not specifically flush the coronary arteries. The brain and cerebellum were also harvested with minimal pre-freezing procedures. We added this precision in the Methods section in our manuscript.

We acknowledge that residual blood may account for the presence of GDF15. However, if this were the case, other tissues such as the brain, the liver and the brain would also show staining for the GDF15 mature protein, which was not observed. This limitation was noted in our manuscript. “Interestingly, highly vascularized organs such as the liver, lungs and muscles, did not show an increase in GDF15 protein expression, eliminating the hypothesis of a direct link between elevated tissue and circulating blood GDF15.” 

5. The use of different animals in the neurological scoring presented in Figures 3, 4, and supplementary Figure 3 raises questions about the consistency of the analysis. Could you provide a rationale for not including all animals in each set of analyses?

For the PCR, given the robustness of the technique, we extracted and analyzed only 8 animals per group to limit costs. For Western Blot, we analyzed all the animals available, i.e. 9 sham, 10 left stroke and 9 right strokes.

Minor Comments:

• The Methods section is overly concise, which makes it challenging to comprehend the exact procedures undertaken, especially concerning the preparation of animals and tissues before WB analysis. A more detailed description would be beneficial.

We apologize for this lack of details in our Methods section. We have revised it accordingly to include the necessary information.

• The manuscript lacks a detailed method for quantifying GDF15 in plasma.

This has also been added, according to the reviewer’s recommendation

• The statistical significance denoted by one to three stars is not explained in the legend or methods. Please ensure that all symbols used in figures are adequately defined.

We thank the reviewer for bringing this to our attention. Clarifications of statistical significance have been added to the figure legends and all figures have been carefully revised.

• The term "very significant" used in the conclusions is ambiguous. To maintain clarity and precision in reporting results, it is advisable to use established terms such as "significant." Please specify which data the term refers to and consider revising it accordingly.

We agree with the reviewer and have removed “very significant” in the conclusion.  

Reviewer #2

The authors express their gratitude to the reviewer for the meticulous review of our manuscript and for the insightful comments provided.

Reviewer #2: The authors have carried out interesting work looking at GDF15 in the context of stroke in a pre-clinical model. This work is interesting, and novel but some additions would be useful to improve the quality of the manuscript. If the authors have brain sections available, it would be useful to complete the analyzes for immunochemistry for GDF15 between the different areas. In addition, if the authors could test the detection of GFRAL in the brain cortex and discuss this results if GRAL present or not could be interesting.

We fully agree with the reviewer’s insightful comment that immunohistochemical staining would provide valuable data for accurately determining the specific brain areas involved inGDF15 expression in both parts of the cortex following stroke, and in other tissues such as the heart. The identification of GFRAL would also be of great interest in brain sections. Unfortunately, all brain and heart samples used for biochemical analysis have been exhausted, and no further experiments can be conducted in the near future. We sincerely apologize for this limitation.

---

## [Decision Letter · Decision Letter 1]

1 Jul 2024

Proximal and distant expression of growth differentiation factor 15 (GDF15) correlate with neurological deficit following experimental ischemic stroke

PONE-D-24-07402R1

Dear Dr. Vergely,

We’re pleased to inform you that your manuscript has been judged scientifically suitable for publication and will be formally accepted for publication once it meets all outstanding technical requirements.

Kind regards,

Clive J. Petry, PhD

Academic Editor

PLOS ONE

Additional Editor Comments (optional):

Reviewers' comments:

Reviewer's Responses to Questions

**Comments to the Author**

1. If the authors have adequately addressed your comments raised in a previous round of review and you feel that this manuscript is now acceptable for publication, you may indicate that here to bypass the “Comments to the Author” section, enter your conflict of interest statement in the “Confidential to Editor” section, and submit your "Accept" recommendation.

Reviewer #2: (No Response)

2. Is the manuscript technically sound, and do the data support the conclusions?

Reviewer #2: Yes

3. Has the statistical analysis been performed appropriately and rigorously? 

Reviewer #2: Yes

4. Have the authors made all data underlying the findings in their manuscript fully available?

Reviewer #2: Yes

5. Is the manuscript presented in an intelligible fashion and written in standard English?

Reviewer #2: Yes

6. Review Comments to the Author

Reviewer #2: I understand the fact that the absence of remaining material prevents me from addressing my comments.

7. PLOS authors have the option to publish the peer review history of their article (what does this mean?). If published, this will include your full peer review and any attached files.

Reviewer #2: No

---

## [Editor Report · Acceptance letter]

4 Jul 2024

PONE-D-24-07402R1 

PLOS ONE

Dear Dr. Vergely, 

I'm pleased to inform you that your manuscript has been deemed suitable for publication in PLOS ONE. Congratulations! Your manuscript is now being handed over to our production team.

Kind regards, 

on behalf of

Dr. Clive J. Petry 

Academic Editor

PLOS ONE